# Phosphatidylethanolamine Deficiency and Triglyceride Overload in Perilesional Cortex Contribute to Non-Goal-Directed Hyperactivity after Traumatic Brain Injury in Mice

**DOI:** 10.3390/biomedicines10040914

**Published:** 2022-04-15

**Authors:** Lisa Hahnefeld, Alexandra Vogel, Robert Gurke, Gerd Geisslinger, Michael K. E. Schäfer, Irmgard Tegeder

**Affiliations:** 1Institute of Clinical Pharmacology, Medical Faculty, Goethe-University, 60590 Frankfurt, Germany; hahnefeld@med.uni-frankfurt.de (L.H.); alexandra-vogel@hotmail.com (A.V.); robert.gurke@itmp.fraunhofer.de (R.G.); geisslinger@em.uni-frankfurt.de (G.G.); 2Fraunhofer Institute for Translational Medicine and Pharmacology ITMP, 60596 Frankfurt, Germany; 3Fraunhofer Cluster of Excellence for Immune Mediated Diseases (CIMD), 60596 Frankfurt, Germany; 4Department of Anesthesiology, University Medical Center, Johannes Gutenberg-University Mainz, 55131 Mainz, Germany; michael.schaefer@unimedizin-mainz.de

**Keywords:** traumatic brain injury, cortical impact, triglycerides, microglia, phosphatidylethanolamines, hyperactivity

## Abstract

Traumatic brain injury (TBI) is often complicated by long-lasting disabilities, including headache, fatigue, insomnia, hyperactivity, and cognitive deficits. In a previous study in mice, we showed that persistent non-goal-directed hyperactivity is a characteristic post-TBI behavior that was associated with low levels of endocannabinoids in the perilesional cortex. We now analyzed lipidome patterns in the brain and plasma in TBI versus sham mice in association with key behavioral parameters and endocannabinoids. Lipidome profiles in the plasma and subcortical ipsilateral and contralateral brain were astonishingly equal in sham and TBI mice, but the ipsilateral perilesional cortex revealed a strong increase in neutral lipids represented by 30 species of triacylglycerols (TGs) of different chain lengths and saturation. The accumulation of TG was localized predominantly to perilesional border cells as revealed by Oil Red O staining. In addition, hexosylceramides (HexCer) and phosphatidylethanolamines (PE and ether-linked PE-O) were reduced. They are precursors of gangliosides and endocannabinoids, respectively. High TG, low HexCer, and low PE/PE-O showed a linear association with non-goal-directed nighttime hyperactivity but not with the loss of avoidance memory. The analyses suggest that TG overload and HexCer and PE deficiencies contributed to behavioral dimensions of post-TBI psychopathology.

## 1. Introduction

Traumatic brain injury (TBI) is a major cause of death and long-lasting disability [1]. Even mild traumatic brain injuries may lead to persistent fatigue, headache, widespread pain, unstable mood, depression, poor sleep, or attention deficits [2]; and survivors of serious TBI often suffer from motor, cognitive, and psychosocial impairments [3]. Brain injury leads to complex time-dependent adaptive processes with ongoing inflammation that lead to cell reorganization and formation of a glial scar [1,4,5,6]. The histopathology and immunopathology have been extensively studied in rodent models of TBI [7,8,9], but it is still not well understood how sequential biochemical and cellular events after the injury and secondary damages [10] contribute to or impact the psychopathology, which is astonishingly mild in mice in relation to the size of injuries in TBI models [11,12,13,14].

We recently showed that TBI in mice leads to long-lasting behavioral abnormalities featured by non-goal-directed nighttime hyperactivity with moderate impairment of avoidance memory. Interestingly, behavioral hyperactivity parameters were associated with reduced concentrations of endocannabinoids (eCBs) in the perilesional cortex and plasma. The lower was the anandamide (AEA) or palmitoylethanolamide (PEA), the higher were the hyperactivity and loss of attention [15]. The results suggested profound alterations of phosphatidylethanolamines and possibly further fatty-acid-derived lipids in the injured brain to an extent that the alterations manifested in the plasma. Previous studies found that enzyme inhibitors that block endocannabinoid breakdown reduce the pathology of TBI in mice [16,17,18,19], suggesting that the observed eCB-associated hyperactivity and attention loss in TBI mice were a (co-)causative biological correlate with clinical relevance supported by few experimental studies showing a therapeutic benefit of PEA in mouse brain or spinal cord injury [20,21,22] and of cannabis in human TBI mortality or post-traumatic stress disorder [23,24,25].

In contrast to endocannabinoids, targeted lipid analyses did not reveal pathologic-behavior-associated changes of sphingolipids, which are highly enriched in the brain relative to the periphery. They are basic building blocks for sphingomyelins and membranes. On the contrary, neutral lipids including triacylglycerols and cholesteryl ester are low in the brain because energy is mostly generated via oxidative glucose metabolism and not via beta-oxidation. Based on these differences between the brain and the periphery, lipid patterns likely reveal even mild barrier dysfunctions or mild disturbances of metabolic features. In agreement with this hypothesis, lipid droplets were observed in glial fibrillary acidic protein (GFAP)-positive astroglial cells in the penumbra of transient brain ischemia in rats [26], and a recent study showed an accumulation of neutral lipids in a cellular model of Parkinson’s disease [27]. Lipid accumulation was interpreted as an early “biomarker” of neurodegenerative diseases that was evoked by overexpression of mutant alpha-synuclein in this cell model [27]. Likewise, accumulation of cholesteryl ester was found in microglia from TREM2-deficient Alzheimer’s disease model mice, suggesting impaired cholesterol metabolism after myelin phagocytosis [28] that likely contributes to proinflammatory features of lipid scavenging cells.

The evidence of lipid pathology in brain diseases and the previously observed associations of “TBI-behavior” with endocannabinoids motivated us to further investigate lipid pathology in post-TBI encephalopathy. The aim was to identify lipids that are deregulated in the TBI brain and/or plasma and are associated with behavioral deficits. Therefore, we employed untargeted unbiased Liquid Chromatography (LC)–Orbitrap Mass Spectrometry (MS) technology to reveal alterations of unexpected lipids and pattern changes in TBI versus sham mice at four sites of the ipsilateral and contralateral brain and in the plasma and associated pathological lipids with behavioral features, which were obtained by long-term post-TBI observations in IntelliCages [15].

## 2. Materials and Methods

### 2.1. Mice

C57BL/6N mice were subjected to a traumatic brain injury (TBI) at the age of 7–9 weeks using the controlled cortical impact (CCI) method [29]. We used female mice to allow for fight-free housing of groups of 16 mice in the IntelliCages (please see below). Sham animals were handled identically in terms of anesthesia and skin incision. As craniotomy contributed to brain damage in our TBI model [30,31], we considered craniotomy as part of the injury. Only slight drilling on the exposed skull surface instead of craniotomy was performed in sham mice to allow comparisons between noninjured and injured brains [32]. Mice were kept in a light- and climate-controlled room at a 12 h dark/light cycle, 22 ± 2 °C, and 50 ± 10% humidity; and they had food and water ad libitum. They were randomly assigned to the sham (*n* = 16) and TBI groups (*n* = 16, 1 dropout during surgery, resulting in *n* = 15) (cohort-2 of [15]), and the experiments and data acquisition were performed in an unbiased PC-based fashion. The experiments were performed in accordance with the “principles of laboratory animal care” (NIH publication No. 86-23, revised 1985) and were approved by the Animal Care and Ethics Committees for Animal Research (Darmstadt, Germany; #FK-1094) and the Landesuntersuchungsamt Rheinland-Pfalz (#G12-1-052). The experiments adhered to the guidelines of the Society of Laboratory Animal Science (GV-SOLAS) and were in line with the European and German regulations for animal research and the ARRIVE guidelines.

### 2.2. Controlled Cortical Impact Model of Traumatic Brain Injury

The controlled cortical impact (CCI) model [29] was used to induce TBI in mice as described [33]. Anesthesia was induced with 4% (*v*/*v*) isoflurane and maintained with 2% (*v*/*v*) isoflurane. For craniotomy, mice were fixed into a stereotactic frame. A 4 mm × 4 mm hole was drilled into the cranium with a handheld dentist’s drill above the right parietal cortex. An electromagnetic-driven impactor (Leica) was positioned above, and the right parietal cortex was impacted with a diameter impactor tip of 3 mm, a velocity of 6 m/s, and a duration of 200 ms. The penetration depth was 1.5 mm. Mice were kept on a feedback-controlled heating pad (Hugo Sachs, March-Hugstetten, Germany) during surgery and maintained in an incubator (Draeger, Lübeck, Germany) for 1.5 h after surgery during recovery. The study encompassed 16 mice per group. One TBI mouse dropped out because of a surgical issue. The time schedule is presented in Figure 1.

### 2.3. IntelliCage Behavior

The time schedule for the IntelliCage experiments and the protocols are described in [15]. TBI and sham mice (cohort-2 [15]) were adapted to the system with free access to every corner, with all doors open, water, and food ad libitum. The free adaptation (FA) was followed by “nosepoke adaptation” (NP), during which mice were required to nosepoke on the door to open it for 5 s and get access to the drinking bottle. In an NP-probability task, the door opened for only 2 s with a probability of 50%. Hence, only every second NP opened the door. Fluctuations of activities during FA and NP were used for analysis of circadian rhythms.

In place avoidance acquisition (PAA) and place avoidance extinction (PAEx), mice learned to avoid one corner, in which a nosepoke at the respective doors triggered an airpuff (PAA, 24 h). In the extinction period (PAEx, 5 d), all corners and sides were free. An LED reminded of the punished corner and was switched on above the respective doors upon corner entry. Mice more or less maintained avoidance of the respective corners showing avoidance memory.

In the “place preference learning” task (PPL, 10 d) mice had to learn to prefer one corner. In this corner, doors opened upon nosepoke, giving access to the drinking bottle on one side of the corner. They had to choose the correct side within the correct corner, which was supported by the LED that was switched on above the correct door upon corner entry. The side switched back and forth between day and night (red and green LED).

Visits in correct/incorrect corners, nosepokes at correct and incorrect doors, and licks were the basic parameters, which were continuously recorded in 1 min intervals and allowed for analyses of activity, circadian rhythms (mesor, amplitude, and acrophase), attention, preferences, efforts, compulsiveness, evenness, repetitiveness, and learning and memory. Appendix A explains IntelliCage parameters that describe the behavioral features and patterns. Time courses of few parameters of the present cohort were shown in [15] (cohort-2).

### 2.4. Plasma and Tissue Collection and Lipidomic Analyses

Mice were euthanized by carbon dioxide and cardiac puncture, whereby blood was collected into K^+^ EDTA tubes, centrifuged at 1300× *g* for 10 min at 4 °C, and the plasma transferred to a fresh tube and snap-frozen in liquid nitrogen. The brain was rapidly removed, and perilesional and sublesional brains were excised from the ipsilateral side, as well as the corresponding tissue from the contralateral side. Equivalent tissue pieces were obtained from sham mice. The tissue was weighed and then snap-frozen in liquid nitrogen. Samples were stored at −80 °C until analysis.

Tissue samples were homogenized prior to lipid extraction using wet grinding in a Precellys 24 (Bertin Instruments, Montigny-le-Bretonneux, France) at 10 °C. The tissue homogenates had a concentration of 0.05 mg/µL in ethanol/water (1:3, *v*/*v*), and 10 zirconium dioxide balls were added for the grinding. Tissue and plasma samples were extracted using methyl tert-butyl ether [34]. For lipid extraction, 150 µL of internal standards (Appendix A) in methanol, 500 µL of MTBE, and 125 µL of 50 mM ammonium formate were added to 20 µL of tissue homogenate or to 20 µL of plasma. The samples were vortexed for 1 min and subsequently centrifuged for 5 min at 20,000× *g* at ambient temperature. The upper organic phase was transferred to a new tube, and 200 µL of a mixture of MTBE/methanol/water (10:3:2.5, *v*/*v*/*v*, upper phase) was added to the lower phase, which was vortexed again for 1 min and centrifuged for 2 min at 20,000× *g* at ambient temperature. Chemicals were from Sigma (Munich, Germany).

The combined upper phases were split into separate aliquots for measurement in positive and negative ion mode and dried under a nitrogen stream at 45 °C before storage at −80 °C. Prior to analysis, samples were dissolved in 120 µL of methanol.

The analysis was performed on an Exploris 480 Orbitrap MS with a Vanquish Horizon (both Thermo Fisher Scientific, Dreieich, Germany) and a Zorbax RRHD Eclipse Plus C8 1.8 µm 50 mm × 2.1 mm ID column (Agilent, Waldbronn, Germany) with a precolumn of the same type. The mobile phases were 10 mM ammonium formate and 0.1% formic acid in water and 0.1% formic acid in acetonitrile/isopropanol (2:3, *v*/*v*) with a flow rate of 0.3 mL/min. The total run time was 14 min. The MS analyses encompassed an MS scan from 180 to 1500 *m/z* at a 240,000 resolution. MS/MS spectra were obtained using Xcalibur 4.4 software with AcquireX deep scan (Thermo Fisher Scientific, Dreieich, Germany). The lipids were identified with a mass error of +/− 3 ppm and a comparison of the isotope ratio and MS/MS spectra [34].

The samples were measured in a randomized order with pooled quality control samples after every 10th injection. Raw data were evaluated using TraceFinder 5.1 and Compound Discoverer 3.1 software, and peak areas were normalized to internal standards of the respective lipid class (AUC/IS) (all Thermo Fisher Scientific, Dreieich, Germany). Internal standards are listed in Appendix A.

### 2.5. Oil Red O Histology

Mice were anesthetized with 4 vol% isoflurane and decapitated; and brains were dissected, immediately frozen in powdered dry ice, and stored at −20 °C. Brains were cut to 12 µm sections using a cryostat (HM 560 Cryostat, Thermo Scientific). Sections were stained with 0.3% (*w*/*v*) Oil Red O (ORO) in 60% isopropanol and counterstained with hematoxylin according to standard procedures. Images of the brain sections were captured using an Axiovert 200 microscope (Zeiss, Oberkochen, Germany) and 4-fold and 20-fold objective lenses (Zeiss, Oberkochen, Germany).

### 2.6. Data Analysis and Statistics

Data were analyzed with SPSS 27, GraphPad Prism 9.02, Origin Pro 2022, and IntelliCage Plus (TSE Systems GmbH, Bad Homburg, Germany) and FlowR (XBehavior, Switzerland) [35] for IntelliCage experiments and ArrayStar (Lasergene 17) for lipidomic analyses. Data are presented as heatmaps and scatter and bubble plots. Bubble sizes represent a related lipid or the behavior as indicated in the respective figure legends. Volcano plots were used to assess fold differences of lipids between sham and TBI mice groups relative to the *p*-values of *t*-tests. Sample sizes are given in the respective figure legends. The study included 16 sham and 15 TBI (one dropout during surgery) mice.

In IntelliCage, the numbers of visits, nosepokes, and licks were analyzed to assess overall activity, compulsiveness, and circadian rhythms. The proportion of errors was used to reveal learning and memory. Canonical discriminant analysis (CanDisc) was used to assess discrimination of the groups and prediction of group membership according to CanDisc scores and to reveal the key behavioral parameters that separated sham versus TBI mice. Candidate behavioral parameters were then submitted to linear regression analyses, which were used to assess the relationship of behavior and lipid or lipid class and linear associations of lipids of different classes. A significant association required that the 95% confidence intervals of the slope of the regression line were different from zero. Principal component analysis (PCA) was used to assess which lipid class or behavioral feature had the strongest impact on variability between groups. Data of individual lipids were analyzed with Student’s *t*-tests (2 groups) or univariate or 2-way analysis of variance (ANOVA) and subsequent post hoc *t*-tests using a Dunnett, or groupwise adjustment of alpha according to Šidák.

## 3. Results

### 3.1. Lipidome Patterns in Brain Tissue and Plasma

We assessed overall lipid patterns in plasma and brain tissue using hierarchical clustering of lipid species with Euclidean distance metrics and visualization as heatmaps (Figure 2 and Appendix A). A total of 309 lipid species were reliably measured in plasma, and 213 in brain tissue, and Figure 2 shows an overview of 213 lipids, which were detected at four brain sites. The red rectangle highlights the deregulated lipid pattern in TBI mice of the ipsilateral cortex. The heatmap also reveals differences between the cortex and subcortex, but is similar in both groups. Figure 2 shows the mice in the sequence of their IDs. In Appendix A, clustering of mice (columns, top dendrogram) and lipids (lines, left dendrogram) revealed that the ipsilateral TBI cortex differed from all other regions.

Changes of lipid species were revealed in volcano plots for each region (Figure 3A–E). Please observe the differences of the scaling of the *y*-axis and *x*-axis of the ipsilateral cortex (highlighted panel in red, Figure 3B). The x-axes show the log2 difference (=fold difference) between groups. Lipids that were increased in TBI are positive, and those that were reduced are negative. The *y*-axis shows the negative logarithm of the *t*-test *p*-value. A 1.5-fold change (log2 difference of ±0.6) plus an FDR adjusted *p*-value (*q*-value) of 0.01 (−log*p* = 2) was considered statistically significant. Volcano plots revealed a major increase in triacylglycerols in the ipsilateral cortex of TBI mice (Figure 3B). TGs are normally low in the brain. Volcano plots also show a reduction of hexosylceramides (HexCer) and phosphatidylethanolamines (candidate PE-O 38:3) in the ipsilateral cortex and an increase in sphingomyelin species (Figure 3C). There were only subtle increases of long-chain TGs in plasma (Figure 3A), but, as a group, still a prominent change.

### 3.2. Interrelationship of Deregulated Lipids

To assess the interdependence of lipids and to what extent they impacted group membership, we used discriminant PCA analysis (Figure 4A) and subsequent regression analyses of lipid versus lipid (Figure 4B–E). PCA biplots show an increase in TG and sphingomyelins (SM) (Figure 4A). High SMs were mostly SMs with C-chains of 33–36 atoms, whereas SMs above 40 carbon atoms were not affected. SMs were not significantly associated with behavioral features (not shown). Scatter clouds to SM versus TG (Figure 4B) show highly different nonoverlapping clusters of TBI versus sham mice.

Hexosylceramides (glucosylceramides and lactosylceramides) were the most strongly reduced lipid class in the ipsilateral TBI cortex, suggesting a defect of globosides and ganglioside biosynthesis [36]. Interestingly, hexosylceramides were linearly associated with PE-O and PS species (Figure 4C,D), pointing to a so far not recognized metabolic pathway that connected these deranged lipids in the TBI brain.

### 3.3. Accumulation of Triacylglycerols in the Perilesional Cortex after TBI

Triacylglycerols were the most strongly upregulated class of lipid species in the ipsilateral cortex of TBI mice. To assess specific regulations of TG species and the relationship to their plasma levels as a putative source, TG species were clustered and visualized as heatmaps (Figure 5). TGs are normally kept low in the brain as compared with plasma, suggesting that a disruption of the blood–brain barrier allowed the influx of plasma lipids that accumulated in perilesional brain regions.

To understand the biology behind the TG overload, TGs were sorted according to chain length and saturation, as shown in Figure 6A. Interestingly, the undulating expression patterns of low and high saturated species were maintained in TBI mice, but the majority of TG species were highly increased in the ipsilateral TBI cortex, with few exceptions, including TG 50:0, 52:0, 54:1, and 56:1, which are unsaturated and monounsaturated TGs.

We asked where these loads of neutral lipids are stored. Oil Red O staining of histological vibratome sections of TBI mice 1 year after the injury revealed highly increased neutral lipids in cells of the perilesional border (Figure 6B), suggesting that these border cells take up TGs from surrounding fluids and are unable to breakdown lipid debris and, therefore, deposit TGs in lipid droplets. These border cells are positive for inflammatory and glia markers [37,38], contribute to ongoing inflammation, and are drivers for the scarring tissue transformation. Weaker, more orangelike ORO staining also occurred in a few cells deeper in the brain tissue.

### 3.4. Reduced Phosphatidylethanolamines in the Ipsilateral Cortex after TBI

We previously showed that low endocannabinoids, anandamide (AEA), palmitoylethanolamide (PEA), and oleoylethanolamide (OEA) are associated with non-goal-directed hyperactivity after TBI. Phosphatidylethanolamines are precursors of these ethanolamide endocannabinoids [39,40,41], and volcano plots revealed a decrease in the ipsilateral cortex after TBI (Figure 3B). We therefore assessed this lipid class in more detail and plotted PE and PE-O (ether-linked PE) levels, sorted for chain lengths as scatter plots (Figure 7A). Several PE and PE-O species were significantly reduced in the ipsilateral TBI cortex, particularly PE-O with a chain length of 18 or 38 C-atoms and PE with 36 C-atoms. There were no differences at other brain sites or in the plasma (Figure 3). Key regulated candidates were linearly associated with PEA, which consists of 16 C-atoms (Figure 7B_1_). Associations of OEA (18 C-atoms) and AEA (20 C-atoms) were weaker and confined to the respective chain lengths or multiples thereof (Figure 7B_2_ and Appendix A). To further reveal the pattern changes of phosphatidylethanolamines, data were transformed to percentages of the mean of sham mice and are depicted as polar plots that clearly reveal the shrinkage of the TBI star of PE, PE-O, and less so LPE species (Figure 7C), and suggest that such a loss of PE, which are highly abundant in the brain, would impact function and manifest in behavioral abnormalities.

### 3.5. Post-TBI Behavior

We previously showed that TBI causes persistent nighttime hyperactivity in association with loss of attention [15]. The behavior manifested in a high frequency of corner visits in IntelliCages without an attempt to get something to drink. It was therefore interpreted as “non-goal-directed nighttime hyperactivity”, and was extracted as a key feature from multiple parameters [15]. Daytime activity did not differ between TBI and sham mice [15]. In the present lipidomics cohort, we confirmed and further analyzed behavioral traits in association with tissue and plasma lipids.

To assess behavioral patterns, behavioral parameters of goal-directed and non-goal-directed activity, exploration, licking, compulsiveness, circadian fluctuations, avoidance learning and memory, spatial learning, and sensory functions were transformed to percentages of the respective median behavior of sham mice and are presented as a polar plot (Figure 8). The plot reveals an increase of nighttime activity (NPVisits/h night, amplitude), licking (Licks/Visit, licking duration), and proportion of errors particularly in a task of avoidance memory. Mice normally maintain avoidance of a corner and door (side) that triggered an airpuff during the acquisition phase of avoidance learning (PAA). A high proportion of errors indicates rapid loss of memory. The full phenotype may be described as “hyperactive, compulsive, and forgetful”. In agreement with the key features, the structure matrix of a canonical discriminant analysis identified nighttime non-goal-directed NPVisits/h, amplitude, proportion of errors in an avoidance extinction task (errors PAEx), and Licks/Visit as the leading parameters, which were therefore selected to assess behavioral associations with lipid deregulations.

### 3.6. Associations of Deregulated Lipids with Abnormal Post-TBI Behavior

Nighttime corner visits with nosepoke but without licks (nighttime NPVisits/h) show TBI-associated hyperactivity. The proportion of errors in avoidance extinction shows low memory. We therefore primarily used these behavioral parameters to assess associations with key lipid classes (Figure 9A–H and Appendix A). Further analyses included compulsiveness (Licks/Visit) and avoidance learning acquisition (errors in PAA) (Figure 9I).

Regression analyses revealed that high TGs were positively associated with hyperactivity (one outlier mouse excluded, Figure 9A). On the contrary, there were significant negative associations for PE, PE-O, and hexosylceramides (HexCer) (i.e., the lower the PE (or PE-O or HexCer) in the ipsilateral cortex was, the stronger was the hyperactivity (Figure 9E–G). The best separation of sham versus TBI clusters was achieved for nighttime NPVisits/h versus HexCer, suggesting that HexCer deficiency had a strong impact on the hyperactivity. HexCers are basic building blocks for globosides and gangliosides.

Sham and TBI clusters also separated in XY-graphs of avoidance “memory” versus key lipid classes (Appendix A). In these comparisons, memory was represented by the proportion of errors during the extinction phase of avoidance learning. TBI mice showed a higher proportion of errors, but there was no significant linear association of memory–behavior with any on the lipid classes (Appendix A).

The analyses revealed that the lipid classes with the strongest impact on behavioral readouts were TG, PE and PE-O, HexCer, and phosphatidylserines (PS) (Figure 9I). The profiles for each mouse showed characteristic behavioral patterns (bubble sizes in Figure 9I) and suggest that high TGs co-occur with low PE, PS, and HexCer in some mice that show the full phenotype. Overall, bubble sizes for hyperactivity, compulsiveness, and errors are bigger and more heterogeneous in TBI mice, reflecting the high interindividual variability of behavioral outcomes of TBI in mice.

## 4. Discussion

Lipidomic analyses of the present study reveal persistent accumulations of neutral lipids in the ipsilateral perilesional cortex after traumatic brain injury in mice that coincide with low phosphatidylethanolamines and hexosylceramides. The Oil Red O histology of neutral lipids revealed bright lipid-laden cells at the perilesional border, suggesting that these lipid scavenging cells were astrocytes and inflammatory cells that demark the lesion and contribute to the formation of the glial scar. Lipid-laden Nile Red positive cells were previously also observed in the penumbra of an ischemic lesion in a stroke model in rats. They were GFAP positive, suggesting an astroglial origin. Further lipid-laden cells occurred in the lesion core and carried the inflammatory maker ED1 [26], hence suggesting that different cell types can contribute to lipid scavenging in an injured brain. Astrocytes take up lipids from neurons via metabolic coupling [42,43] requiring functional apolipoprotein E (APOE) [44,45]. Mostly, this mechanism is considered to be neuroprotective [46], but lipid accumulation in stressed astrocytes may kill neurons and trigger microglia activation via release of saturated fatty acids [47,48].

TG levels were associated with the extent of the behavioral abnormality, which was featured by nighttime non-goal-directed hyperactivity. TGs were higher in mice with strong hyperactivity represented by nighttime “corner visits with nosepokes but without licks” (NPVisits/h), although one outlier mouse did not fit to this scheme. TGs were, however, not directly associated with errors in a task testing airpuff-based avoidance memory. The proportion of errors was higher in TBI mice, but the memory of the unpleasant experience of an airpuff was not influenced by TG or any other of the lipid classes. It is of note that lipidomic patterns of the contralateral cortex, subcortical brain tissue, and plasma were remarkably alike in TBI and sham mice and that reward-based learning and memory were not negatively affected in TBI mice irrespective of the large cortical trauma. TBI mice also did not show behavior indicative of nociceptive hypersensitivity or disruptions of circadian rhythms [15], suggesting that TBI mice adapted astonishingly well to a unilateral cortical damage at the behavioral level, which stands in contrast to severe human disabilities after TBI.

It is a limitation that our study was performed in mice, and it is unknown whether similar lipid changes were to occur in the human brain after trauma and might depend on ages [49]. In mice, TBI-associated deregulation of PE and TG in the ipsilateral brain did not manifest in significant alterations of the respective plasma lipids, limiting the translational diagnostic value. However, we analyzed whole plasma and did not separate brain-derived exosomes, which were recently shown to reveal post-TBI-associated biomarkers [50]. The localization of the ORO-positive lipid-laden cells suggests that lipidomes of glia-derived exosomes might reveal lipid pathology after brain trauma, supported by plasma biomarker studies that have revealed deregulated miRNAs in exosomes [51,52,53]. It is of note that a subset of miRNAs regulates triglyceride homeostasis [54].

Although TBI mice differ substantially from a human TBI, it is an advantage under ARRIVE considerations that mice show subtle injury phenotypes after TBI, hence justifying the use of a controlled cortical impact as a model. Indeed, the multifactorial reanalysis of individual behavioral data in association with the individual’s lipidome agrees with the 3R principles, particularly the principle of reduction. In this analysis, each mouse was considered an individual “patient” in analogy to human studies. The approach uncovered candidate PE, HexCer, and TG pathways that might be amenable to therapeutic interventions supported by experimental studies showing therapeutic anti-inflammatory effects of the endocannabinoid, PEA in stroke or brain trauma [20,21,22,55], of fatty acid amide hydrolase inhibitors [17,19] and of GM1 gangliosides in TBI mouse models [56,57,58].

TGs are neutral lipid stores consisting of saturated to poly-unsaturated fatty acids that fuel mitochondrial beta-oxidation. Fatty acids are basic building blocks for other structural and signaling lipids. They are generated de novo via fatty acid synthase (FASN) [59,60], which is expressed by astrocytes but not neurons [61]. TGs are broken down by triglyceride lipases and recycled via lipophagy from lysosomal content [62,63,64]. TGs are also taken up from neurons and extracellular fluids via fatty acid transporters [46]. TGs are then stored in lipid droplets that undergo lipophagy on demand for reutilization [65,66] or are expelled via lysosomal exocytosis [67]. The observed intense ORO staining of TBI border cells suggests that reutilization of scavenged TGs or lipolysis was defective or overloaded in these cells, which are likely astroglia and inflammatory cells. Indeed, high digestion of myelin debris has been suggested to account for ORO-positive cells in post-TBI histology [68]. These cells deposited neutral cholesteryl ester and triglycerides upon ingestion of opsonized myelin and contributed to ongoing inflammation [69]. On the other hand, astrocyte-derived lipids are critical for myelination and synapse development [61,70].

TGs are normally low in the brain because neurons generate energy via glycolysis and oxidative phosphorylation rather than via beta-oxidation, but astrocytes use beta-oxidation to provide energy for neurons [71], and some triglyceride lipases (TGLs) were found in the brain particularly at border sites, including adipose triglyceride lipase (ATGL/PNAPLA2) and hormone-sensitive lipase (LIPE) [72,73]. The latter was recently localized at synapses, and knockout of LIPE impaired memory in mice. It has not yet been studied whether specific lipases are deregulated in the TBI brain, but high TGs likely interfere with the removal of protein or organelle wastes, because lipophagy combats with other autophagy-dependent degradation pathways [74]. Indeed, lipid droplets were found to impair the removal of damaged mitochondria via mitophagy in a cellular neurotoxicity model [75]. High TG levels in the brain may therefore increase the risk of waste accumulation and show that the brain barrier towards blood and CSF is leaky, allowing the influx of plasma TG and cholesterol and failure to degrade such high amounts of neutral lipids.

The TG burden was particularly high (ORO stain) in border cells, whereas deeper brain layers yielded light orange tinging in agreement with the “normal” lipid profile of the subcortical brain tissue. The accumulation of TG in the TBI cortex was not specific for a certain chain length or saturation. With a few exceptions, all TG species were increased in the perilesional TBI cortex and were linearly associated with each other. In contrast, there was no linear (or nonlinear) association of TG with any other class of lipids, and TG represented an independent factor in multivariate analyses. Nonoverlapping separation of TBI versus sham mice was achieved by XY scatter plots of TG versus sphingomyelins, or versus PE or HexCer, which were both reduced in the ipsilateral TBI cortex. Based on ORO staining and current knowledge, it is likely that TG overload occurs mainly in glial cells, whereas the lowering of PE and HexCer reflects a neuronal shortage, which needs to be addressed further in cell-type-specific lipidome studies [76].

Phosphatidylethanolamines are precursor lipids for the ethanolamide endocannabinoids, AEA, OEA, and PEA [39,41,77], which we identified as biological correlates of TBI-associated hyperactivity in our previous study [15], and are supposed to trigger anti-inflammatory signals via activation of canonical and noncanonical cannabinoid receptors [78,79,80,81,82]. In agreement with the endocannabinoid hypothesis, PE and PE-O levels were linearly associated with eCB, particularly with PEA (Figure 6B), and indeed, low PE or PE-O levels were associated with strong hyperactivity. Hence, the lipidomic results of the present study confirm the loss of phosphatidylethanolamines and strengthen the idea that they are predictors of behavioral abnormalities of hyperactivity. The metabolic deficiency likely results in a shortage of endocannabinoids, particularly PEA. In humans, low levels of circulating ethanolamide endocannabinoids were associated with intrusive memories in the context of post-traumatic stress disorder [83,84] and attention deficit hyperactivity disorder [85,86], suggesting a translational clinical relevance of the observed PE and PE-O deficiency, also supported by therapeutic effects of PEA in brain or spinal cord trauma models [20,22,87,88].

Hexosylceramides are glucosylceramides or lactosylceramides. They are metabolites of ceramides and precursors of gangliosides, respectively. High levels of glucosylceramides have been associated with sporadic Parkinson’s disease [89,90,91] and neurologic forms of Gaucher disease [92,93,94] that result from heterozygous or homozygous “loss-of-function” mutations of the GlcCer-degrading enzyme, acidic lysosomal glucocerebrosidase (GBA1). GBA1 dysfunctions increase the toxicity of alpha-synuclein, which in turn impairs vesicle biology and lysosomal functions [95,96,97]. Hence, elevated levels of GlcCer rather than low levels prevail in lysosomal disorders and neurodegenerative diseases [89]. Instead, low HexCer levels in the TBI cortex point to a deficiency of lactosyl- and galactosylceramides, which are required for building complex lipids, such as globosides and gangliosides [36]. In agreement, treatment of mice with GM1 ganglioside has improved the outcome in mice TBI models [56,57,58], and gangliosides are currently reconsidered putative therapeutics for neurodegenerative diseases if poor brain penetration were pharmaceutically solved [98]. It is a limitation that lipidomic mass spectrograms did not differentiate lactosyl- and hexosylceramides (both C6 sugars), and that complex structures of globosides and gangliosides were lost during extraction procedures.

In summary, lipidomics revealed an overload of neutral lipids in an injured TBI cortex likely resulting from barrier defects, influx of plasma lipids, defective lipolysis, and deposition of lipid debris. Lipid pathology also manifested in a deficiency of hexosylceramides and phosphatidylethanolamines in association with low endocannabinoids. At the behavioral level, lipid deregulations were associated with nighttime non-goal-directed hyperactivity but not with a loss of avoidance memory, showing differential lipid vulnerability of cognitive dimensions in TBI mice.

## Figures and Tables

**Figure 1 biomedicines-10-00914-f001:**
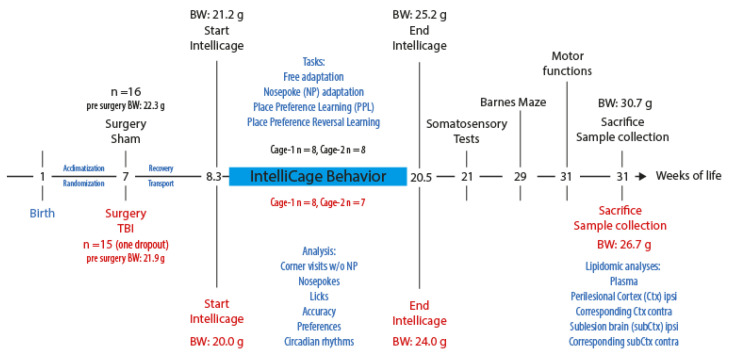
Time schedule of the behavioral experiments and plasma and tissue collection (cohort-2 of [15]). Abbreviations: BW, body weight; g, grams; *w*/*o*, with and without.

**Figure 2 biomedicines-10-00914-f002:**
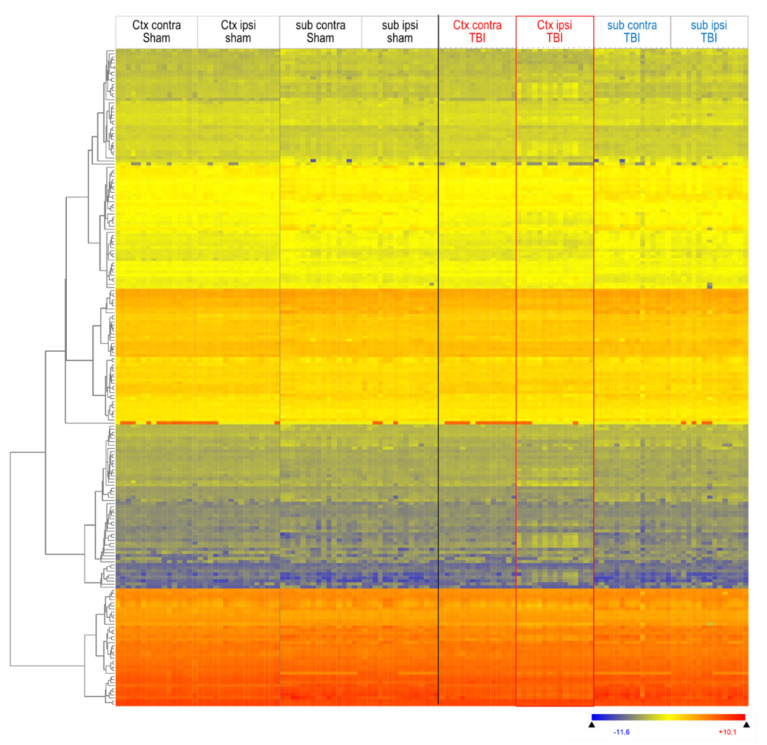
Heatmap of lipid species in ipsilateral and contralateral cortical and subcortical brain tissue 5.6 months after TBI versus sham surgery. Lipids were submitted to hierarchical clustering using Euclidean squared distance metrics. Each horizontal line is a lipid species; each vertical line is a mouse at one site (*n* = 16 sham, *n* = 15 TBI mice). Mice are depicted in the sequence of their IDs. The left dendrogram shows the lipid clusters. The red rectangle highlights the deregulated group of TBI Ctx ipsilateral. The color scale ranges from −3 to +3 SD.

**Figure 3 biomedicines-10-00914-f003:**
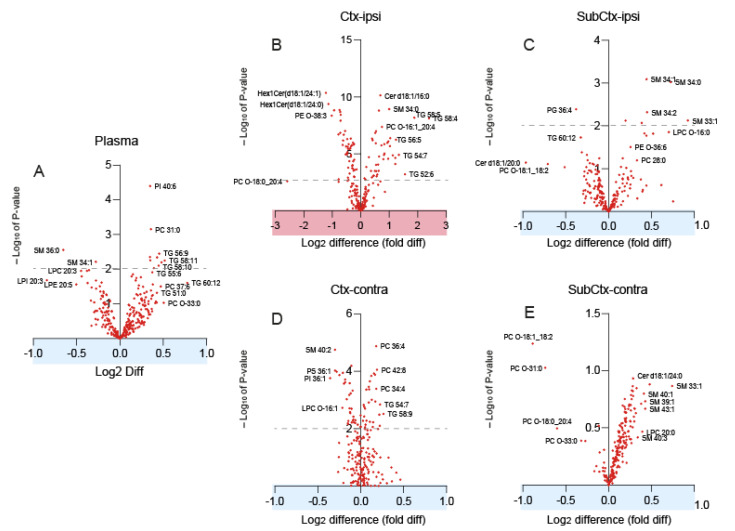
Volcano plots of lipids in the plasma, ipsi-, and contralateral cortex and subcortex. The x-axes show the log2 difference between groups consisting in *n* = 16 sham and *n* = 15 TBI mice. Lipids that were increased in TBI are positive; those that were reduced are negative. The *y*-axis shows the negative logarithm of the *t*-test *p*-value. The dashed line indicates a *p*-value of 0.01. (**A**) Plasma. (**B**) Ctx-ipsi, ipsilateral perilesional cortex. (**C**) SubCtx-ipsi, ipsilateral subcortical brain. (**D**) Ctx-contra, contralateral cortex. (**E**) SubCtx-contra, contralateral subcortical brain. Abbreviations: TG, triacylglycerol; PC, phosphatidylcholine; SM, sphingomyelin; Cer, ceramide; LPC, lysophosphatidylcholine; PI, phosphatidylinositol; PS, phosphatidylserine.

**Figure 4 biomedicines-10-00914-f004:**
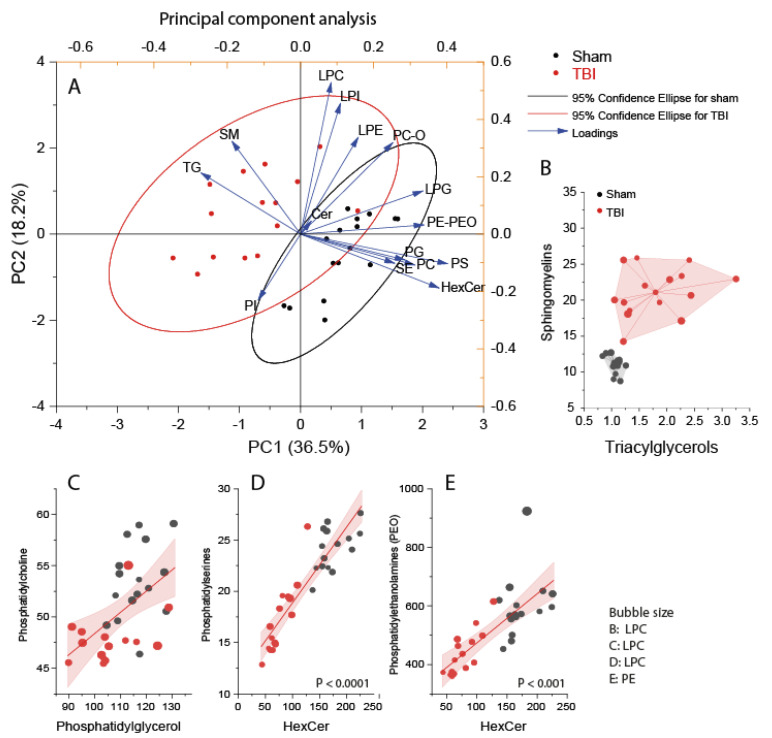
Principal component analysis and association of lipid classes in the ipsilateral perilesional cortex of sham (*n* = 16, black) and TBI (*n* = 15, red) mice. Lipid species of different chain lengths and saturation were summed per lipid class. Scatters represent individual mice. (**A**) PCA component PC1 versus PC2. Asterisks show the loading. Circles are 95% CI. (**B**) Triacylglycerols versus sphingomyelins. Bubble size LPC. (**C**) Linear regression fit with 95% CI of PG versus PC. Bubble size LPC. (**D**) Linear regression of HexCer versus PS, bubble size LPC. (**E**) HexCer versus PE-O, bubble size PE. Abbreviations: TG, triacylglycerols; PE-O, phosphatidylethanolamines with ether-linkage; Cer, ceramides; HexCer, hexosylceramides; SM, sphingomyelin; SE, sterol ester; PC, phosphatidylcholines; LPC, lysoPC. PS, phosphatidylserine; PG, phosphatidylglycerol; LPG, lysoPG. Lipid units are AUC/IS (i.e., the ratio of the area under the curve of the analyte and the AUC of the internal standard).

**Figure 5 biomedicines-10-00914-f005:**
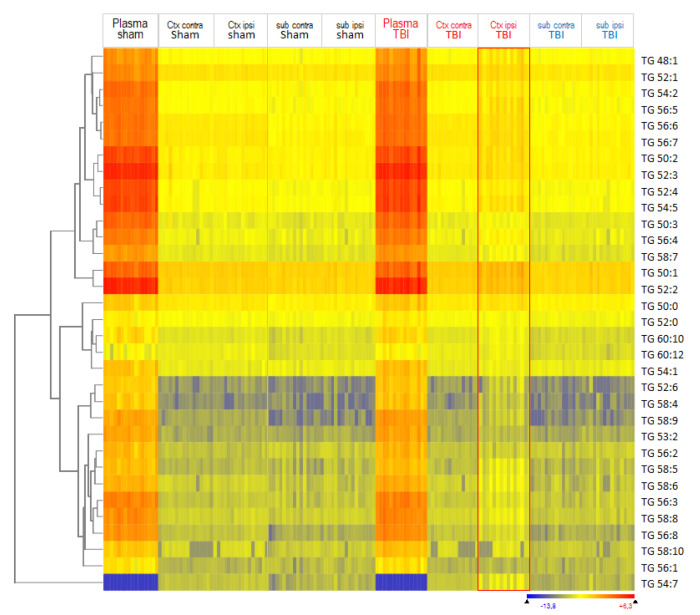
Heatmap of the triacylglycerols in the plasma and ipsilateral and contralateral cortical and subcortical brain 5.6 months after TBI or sham surgery. Each horizontal line is a TG species; each vertical color line is one mouse at one site, each from left to right in the sequence of the mouse IDs (*n* = 16 sham, *n* = 15 TBI). The group blocks are indicated by lines between headers. The red rectangle highlights the ipsilateral TBI cortex. The left dendrogram shows the hierarchic clustering of TG according to concentrations using Euclidean distance metrics. The color scale ranges from −3 to +3 SD. Abbreviations: Ctx, cortex; Sub, subcortical brain tissue; TBI, traumatic brain injury; TG, triacylglycerol.

**Figure 6 biomedicines-10-00914-f006:**
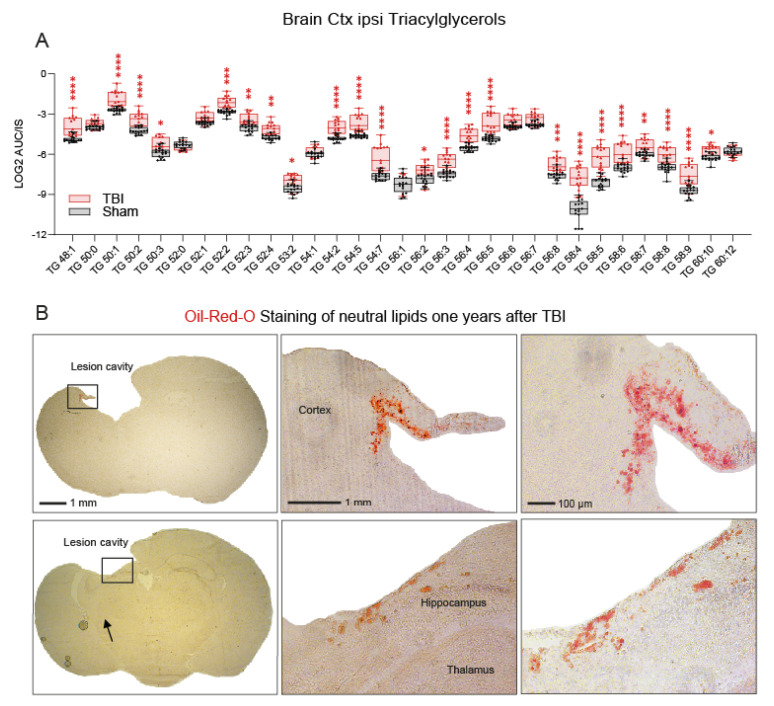
Accumulation of triacylglycerols in the ipsilateral cortex after TBI. (**A**) Box/scatter plot of triacylglycerols of different chain lengths and saturation 5.6 months after the injury. The scatters represent individual mice, *n* = 16 sham and *n* = 15 TBI. The box shows the interquartile range; the whiskers range from minimum to maximum. The data were compared via two-way ANOVA for the TG X group (sham versus TBI) and subsequent post hoc analysis according to Šidák. * *p* < 0.05, ** *p* < 0.01, *** *p* < 0.001, **** *p* < 0.0001. The unit AUC/IS is the ratio of the area under the curve of the analyte of the mass spectrogram and the AUC of the internal standard (IS is listed in Appendix A). (**B**) Oil Red O staining of neutral lipids in cryosections of TBI mice 12 months after the injury. The images show examples of brain sections at two different Bregma levels (−0.34 and −1.94 mm; according to the Paxinos and Franklin’s mouse brain atlas) from two different animals in order to illustrate that Oil Red O stains cells are consistently distributed at the border across the brain lesion irrespective of the Bregma position. The differences in lesion size are due to the different Bregma levels. The upper lesion cavity appears to be larger owing to a fusion with the lateral ventricle at Bregma −0.34. The left panel shows the overview; the right panels show high magnification views of the areas indicated by rectangles. Scale bars: 1 mm, 100 µm.

**Figure 7 biomedicines-10-00914-f007:**
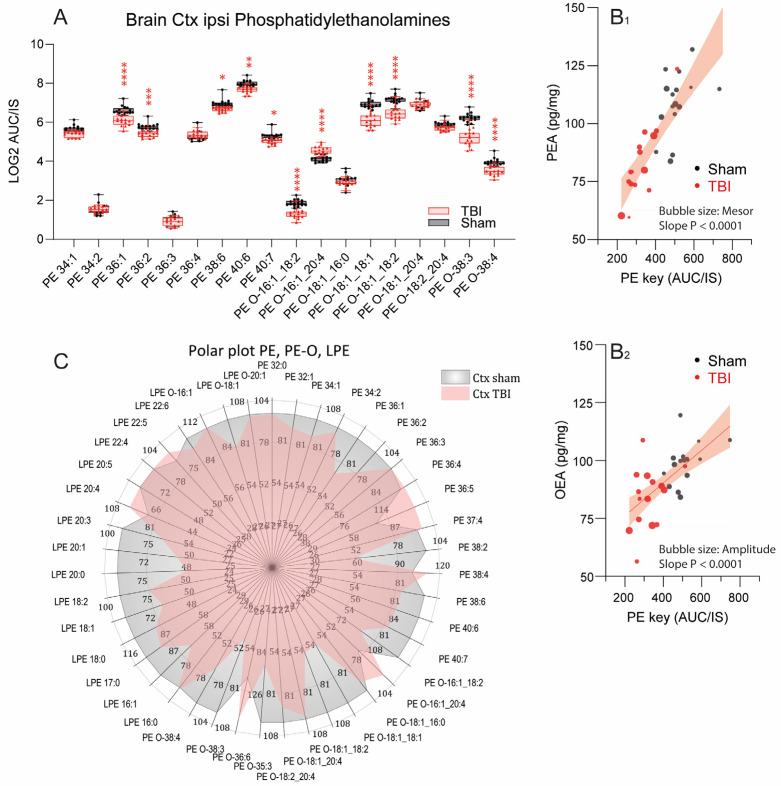
Scatter and polar plots of phosphatidylethanolamines in the ipsilateral cortex 5.6 months after TBI. (**A**) Scatter plot of phosphatidylethanolamines (PE and PE-O species) of different chain lengths and saturation, *n* = 16 sham mice and *n* = 15 TBI mice. Each scatter is a mouse. The data were compared via two-way ANOVA for PE (within subject factor) by group (sham versus TBI, between subject factors) and subsequent post hoc analysis according to Šidák. * *p* < 0.05, ** *p* < 0.01, *** *p* < 0.001, **** *p* < 0.0001. (**B**) Association of phosphatidylethanolamines (key regulated PE and PE-O species) in the perilesional ipsilateral cortex with the endocannabinoids, palmitoylethanolamide (PEA, **B_1_**), and oleoylethanolamide (OEA, **B_2_**). The line shows the linear regression line; the shaded area is the 90% confidence interval. The slopes differed significantly from zero. AUC/IS: ratio of the area under the curve of the analyte and the AUC of the internal standard. Bubble sizes reveal behavioral circadian parameters indicative of median activity (mesor) and day-to-night fluctuations (amplitude). (**C**) Polar plots show median percentages of LPE, PE, and PE-O species in the ipsilateral cortex of sham and TBI mice, normalized to the mean of sham mice.

**Figure 8 biomedicines-10-00914-f008:**
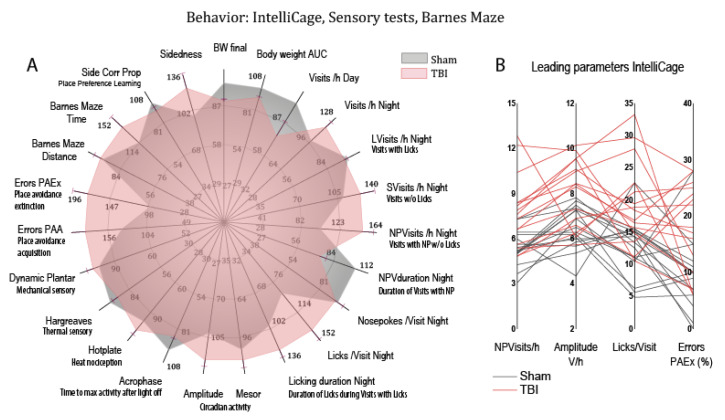
Behavior of sham (*n* = 16, grey) versus TBI (*n* = 15, red) mice. Behavioral data were obtained in different tasks in IntelliCages (PAA (place avoidance acquisition), PAEx (place avoidance extinction), PPL (place preference learning), sidedness, circadian mesor, amplitude, acrophase), spatial learning on a Barnes maze, and sensory tests of thermal and mechanical perception and nociception (hotplate, Hargreaves, dynamic von Frey). (**A**) The polar plot shows multiple dimensions of behavior. To allow for a combined analysis of different readouts/units, raw data were transformed into percentages of the median of sham mice, and mean percentages were plotted on radial y-axes. Error bars are SEM of TBI mice. (**B**) Multiple line plots representing individual mice of four candidate behavioral parameters that contributed most to the discrimination between groups.

**Figure 9 biomedicines-10-00914-f009:**
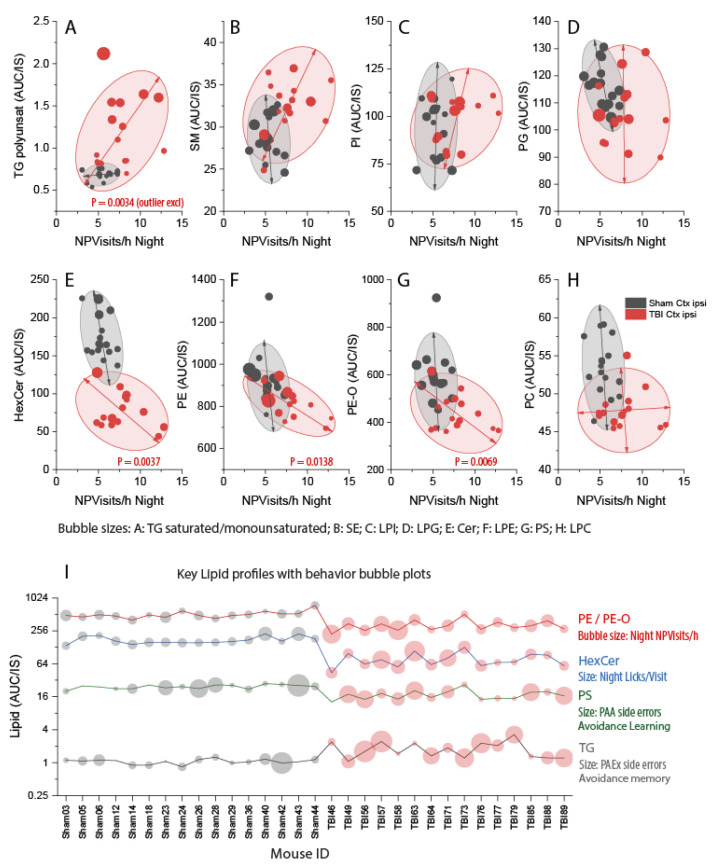
Associations of non-goal-directed hyperactivity (*x*-axis) with lipid classes (*y*-axis) in the ipsilateral perilesional cortex of sham (*n* = 16, black) versus TBI (*n* = 15, red) mice. Non-goal-directed hyperactivity is represented by IntelliCage nighttime visits with nosepokes but without licks per hour (NPVisits/h) (*X*-axis). NPVisits/h were recorded daily, and the mouse’s nighttime average was used. NPVisits/h were associated with lipid classes on the *y*-axis. The bubble size is given by the respective lyso-form or by a related lipid. AUC/IS (unit) is the ratio of the area under the curve of the analyte and the AUC of the internal standard. Individual lipid species of different chain lengths and saturation were summed per lipid class. Scatters represent individual mice. The ellipses show the 85% CI of the prediction (outliers excluded). *P*-Values show statistically significant slopes of regression lines of TBI mice. (**A**–**H**) NPVisits/h versus lipids as indicated (*y*-axis). Bubble sizes as indicated. Abbreviations: TG, triacylglycerol; SM, sphingomyelin; SE, sterol ester; PI, phosphatidylinositol; LPI, lyso-PI; PG, phosphatidylglycerol; LPG, lyso-PG; HexCer, hexosylceramides; Cer, ceramides; PE, phosphatidylethanolamines; LPE, lyso-PE; PE-O, phosphatidylethanolamines with ether linkage; PS, phosphatidylserine; PC, phosphatidylcholine; LPC, lyso-PC. (**I**) Profile plot of key regulated PE, HexCer, PS, and TG per mouse. Bubble sizes show behavioral parameters as indicated. PAA, place avoidance acquisition; PAEx, place avoidance extinction.

## Data Availability

The data generated for this manuscript are presented within the manuscript or in Appendix A. Raw lipidomic data are available on reasonable request.

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
