# Peer review of "Phosphatidylethanolamine Deficiency and Triglyceride Overload in Perilesional Cortex Contribute to Non-Goal-Directed Hyperactivity after Traumatic Brain Injury in Mice"

_biomedicines, 2022, doi:10.3390/biomedicines10040914_

Round 1
Reviewer 1 Report
Evidence of lipid pathology in brain diseases had been associated with "TBI-behavior" and endocannabinoids. Therefore, the aim of the study was to identify lipids that are deregulated in TBI brain and/or plasma and are associated with behavioral deficits. To this end, the authors employed untargeted unbiased LC-Orbitrap-MS technology to reveal alterations in lipids and pattern changes in TBI versus sham mice at four sites of ipsilateral and contralateral brain and in plasma. Further, the authors attempted to associate pathological lipids with behavioral features, which were obtained by long-term post-TBI observations in IntelliCages.
This is a good study. However, at the DISCUSSION, I missed literature details of lipid metabolism in astroglia which, in my opinion is the main contributor to lipid changes following TBI. Second, since astroglia activity is greatly increased after brain injury with increasing age, the authors shall discuss and clearly state that the use of young animals was a study limitation (see, DOI: 10.2174/156720206775541732). Third, a totally neglected factor in studying behavior is the impact of circadian activity. At what time of the day did authors test the animals (see, DOI: 10.1016/J.PNPBP.2016.01.006).
Author Response
Evidence of lipid pathology in brain diseases had been associated with "TBI-behavior" and endocannabinoids. Therefore, the aim of the study was to identify lipids that are deregulated in TBI brain and/or plasma and are associated with behavioral deficits. To this end, the authors employed untargeted unbiased LC-Orbitrap-MS technology to reveal alterations in lipids and pattern changes in TBI versus sham mice at four sites of ipsilateral and contralateral brain and in plasma. Further, the authors attempted to associate pathological lipids with behavioral features, which were obtained by long-term post-TBI observations in IntelliCages.
We thank the reviewer for evaluation of our manuscript and giving us the insightful comments.
We have added the references as suggested.
The changes in the manuscript are highlighted with WORD's track changes. Yellow marked text is rewritten on Editorial request to avoid similarities with own previous work.
This is a good study. However, at the DISCUSSION, I missed literature details of lipid metabolism in astroglia which, in my opinion is the main contributor to lipid changes following TBI.
The Oil-Red-O positive cells along the injury border have astroglial and inflammatory markers. In stroke, both GFAP positive and ED1 positive cells were stained with Nile Red which is similar to Oil-Red-O [1]. Hence, scavenging cells are likely astroglia and inflammatory cells. The reference is now added, and we have added a discussion of neuron-to-astroglia transfer of lipids and lipid droplets in astrocytes. A number of studies show lipid droplets in astroglia which are considered to beneficial energy stores in some studies but pro-inflammatory in others (Discussion Line 455 pp).
Triglyceride metabolism has not been specifically studied in astroglial cells. One paper reports the expression of adipose triglyceride lipase (ATGL/PNPLA2) in the brain particularly at the brain to CSF interface and at the blood brain barrier [2]. A very recent paper detected expression of hormone sensitive lipase (LIPE) at synapses, and knockout of LIPE impaired memory [3]. We have added a discussion of ATGL and LIPE in the DISCUSSION (Line 492 pp).
Second, since astroglia activity is greatly increased after brain injury with increasing age, the authors shall discuss and clearly state that the use of young animals was a study limitation (see, DOI: 10.2174/156720206775541732).
Traumatic brain injury is very frequent in young people. We therefore think that it was reasonable to induce TBI in young adult mice and observe them up to six months. We do not see that the chosen ages are a limitation of the present study, but have briefly added the age-aspect and the reference in the discussion as suggested (Line 470).
Third, a totally neglected factor in studying behavior is the impact of circadian activity. At what time of the day did authors test the animals (see, DOI: 10.1016/J.PNPBP.2016.01.006).
We agree that circadian aspects are important to consider. Therefore, we used the IntelliCages which provide 24h observation in groups of 16 mice without observer handling. Activity is monitored around the clock, day and night. We have reported the circadian rhythms and day-time dependent behavior in detail previously [4]. The most prominent feature was "non-goal directed nighttime hyperactivity", which is used a key parameter for the present study (Figure 7, 8, 9). Daytime activity did not differ between TBI and sham mice. This information is now added (Line 368). We have also assessed Mesor, Amplitude and Acrophase (Figure 7 and 8, bubble size in Figure 7, Amplitude in Figure 8b). We have added the reference about attention deficit disorder. Line 540.
References
- Kamada, H.; Sato, K.; Iwai, M.; Ohta, K.; Nagano, I.; Shoji, M.; Abe, K. Changes of free cholesterol and neutral lipids after transient focal brain ischemia in rats. Acta Neurochir. Suppl. 2003, 86, 177-180.
- Etschmaier, K.; Becker, T.; Eichmann, T.O.; Schweinzer, C.; Scholler, M.; Tam-Amersdorfer, C.; Poeckl, M.; Schuligoi, R.; Kober, A.; Chirackal Manavalan, A.P., et al. Adipose triglyceride lipase affects triacylglycerol metabolism at brain barriers. J. Neurochem. 2011, 119, 1016-1028.
- Skoug, C.; Holm, C.; Duarte, J.M.N. Hormone-sensitive lipase is localized at synapses and is necessary for normal memory functioning in mice. J. Lipid Res. 2022, 100195.
- Vogel, A.; Wilken-Schmitz, A.; Hummel, R.; Lang, M.; Gurke, R.; Schreiber, Y.; Schäfer, M.K.E.; Tegeder, I. Low brain endocannabinoids associated with persistent non-goal directed nighttime hyperactivity after traumatic brain injury in mice. Sci. Rep. 2020, 10, 14929.
Reviewer 2 Report
Review of Biomedicines article “Phosphatidylethanolamine deficiency and triglyceride overload in perilesional cortex contribute to non-goal directed hyperactivity after traumatic brain injury in mice”
The manuscript Phosphatidylethanolamine deficiency and triglyceride overload in perilesional cortex contribute to non-goal directed hyperactivity after traumatic brain injury in mice” provides a lipidome profile analysis in brain and plasma following traumatic brain injury (TBI) in mice associated with non-goal directed hyperactivity. The manuscript is overall clear and well-written. There are study design and analyses issues that the authors need to clarify and discuss, in addition to a few minor editorial errors.
Major points:
- In the Materials and Methods section, the authors describe that sham/control animals were subjected to anesthesia and skin incision, but that no craniotomy was produced in the parietal bone as was done in the TBI animals. We have seen in both our rat and ferret models of TBI that the craniotomy alone, without any additional injury to the brain, is associated with blood-brain barrier disruption and changes in central and peripheral inflammatory markers. The fact the control animals did not have a craniotomy in this study is a confounding variable that the authors need to address. First the authors should explain in the Materials and Methods section why they opted to not produce a craniotomy in the control animals. Then the authors need to address in the discussion how having the additional skull “injury” in the TBI animals but not the controls may have affected the study findings.
- Results: Figure 6B- the coronal sections of the brain show hugely different lesion sizes in the two sections the authors show. Why is that? This discrepancy in 2 out of the 3 mice is very concerning. I know the authors provide the parameters of the CCI injury but some explanation around the consistency of the injury is needed given the dramatically different lesion sizes shown in this figure. Also, the two sections look like they are from different parts of the brain. this also needs needs to be clarified by the authors and included in the figure legend.
- Discussion- line lines 489-491: could the increase in TG levels be also attributed to a reduction in the cells capacity to degrade TG due to injury? I ask because in the case of neurodegenerative proteins such as beta amyloid, clearance mechanisms of the peptide are impaired after injury and may account for increased levels of the protein in the brain. Is this a possibility in the case of TG? The discussion could benefit from addressing this point.
Minor points:
- Line 86: “consisting in anesthesia, stereotactic mounting,…” should be “consisting of…”
- Line 107-108: please explain why one animal was “dropped out.” Was it due to deviation from the protocol, a surgical issue, illness…?
- Line 140: use euthanized instead of sacrificed.
- Line 141: what temperature was the blood centrifuged at?
- Line 210: spelling error- group-vise should be group-wise.
- Line 234 and henceforward: please be consistent in whether you use capitalization or not for Y- and X-axis. The study has them as both lower and upper case (e.g., x-axis and X-axis). Please be consistent. There is no reason I am aware of for capitalizing x or the y.
- Line 324-326: first sentence needs a reference.
- Line 371-372: I think the authors mean forgetful not forgettable. Please change.
- Line 436: can the authors explain what they mean by “environmental tissue”?
- Line 463: add the word “injury” before phenotype.
Author Response
Review of Biomedicines article “Phosphatidylethanolamine deficiency and triglyceride overload in perilesional cortex contribute to non-goal directed hyperactivity after traumatic brain injury in mice”
The manuscript Phosphatidylethanolamine deficiency and triglyceride overload in perilesional cortex contribute to non-goal directed hyperactivity after traumatic brain injury in mice” provides a lipidome profile analysis in brain and plasma following traumatic brain injury (TBI) in mice associated with non-goal directed hyperactivity. The manuscript is overall clear and well-written. There are study design and analyses issues that the authors need to clarify and discuss, in addition to a few minor editorial errors.
We thank the reviewer for reading of our manuscript, the overall positive evaluation and criticisms that helped us to improve the manuscript. Please find our responses below.
The changes in the manuscript are highlighted with WORD's track changes. Yellow marked text is rewritten on Editorial request to avoid similarities with own previous work.
Major points:
In the Materials and Methods section, the authors describe that sham/control animals were subjected to anesthesia and skin incision, but that no craniotomy was produced in the parietal bone as was done in the TBI animals. We have seen in both our rat and ferret models of TBI that the craniotomy alone, without any additional injury to the brain, is associated with blood-brain barrier disruption and changes in central and peripheral inflammatory markers. The fact the control animals did not have a craniotomy in this study is a confounding variable that the authors need to address. First the authors should explain in the Materials and Methods section why they opted to not produce a craniotomy in the control animals.
We thank you for your insight and we agree that craniotomy per se may cause postsurgical inflammation and headache. We apologize that we have not made this point clear. As correctly stated by this reviewer, craniotomy contributes to brain damage in our TBI model [1]. Craniotomy alone may produce injuries to the dura mater and bleeding from meningeal blood vessels. Therefore, we considered craniotomy as part of the injury in our open skull TBI model to allow for comparisons between non-injured and injured brains.
We now write: "Sham animals were handled identically in terms of anaesthesia and skin incision. As craniotomy contributes to brain damage in our TBI model [1], we considered craniotomy as part of the injury. Only slight drilling on the exposed skull surface instead of craniotomy was performed in sham mice to allow for comparisons between non-injured and injured brains [2].
Then the authors need to address in the discussion how having the additional skull “injury” in the TBI animals but not the controls may have affected the study findings.
As outlined above, we considered craniotomy as part of the injury in our open skull TBI model and believe that the revised information in the Materials and Methods section (Line 88-92) sufficiently explains our experimental approach to compare between non-injured and injured brains.
The behavioural studies started only beyond the critical postsurgical period, and the tissue was obtained 5.5 months after TBI. It appears unlikely that an alternative sham procedure would have masked the differences in phenotypes or lipid profiles.
Results: Figure 6B- the coronal sections of the brain show hugely different lesion sizes in the two sections the authors show. Why is that? This discrepancy in 2 out of the 3 mice is very concerning. I know the authors provide the parameters of the CCI injury but some explanation around the consistency of the injury is needed given the dramatically different lesion sizes shown in this figure. Also, the two sections look like they are from different parts of the brain. this also needs needs to be clarified by the authors and included in the figure legend.
The presented images show brain sections at two different Bregma levels (-0.34 mm and -1.94 mm; according to the mouse atlas from Paxinos et al.) from two different animals in order to illustrate that Oil-Red-O stained cells are consistently distributed along the border across the brain lesion rather than being a single event at a specific region in individual animals. The differences in lesion size reflect the different Bregma levels because the size of the lesion cavity varies with respect to the distance from the primary injury site. In the upper image, the lesion appears to be larger owing to a fusion of the lesion cavity with the lateral ventricle at Bregma -0.34 mm. We have now added an explanation in the figure legend. Photographs of the brains of the "behaviour & lipid animals" were presented in our previous paper [3].
Discussion- line lines 489-491: could the increase in TG levels be also attributed to a reduction in the cells capacity to degrade TG due to injury? I ask because in the case of neurodegenerative proteins such as beta amyloid, clearance mechanisms of the peptide are impaired after injury and may account for increased levels of the protein in the brain. Is this a possibility in the case of TG? The discussion could benefit from addressing this point.
Indeed, the capacity to degrade TG may be impaired after TBI, but the functions of triglyceride lipases in the brain after TBI are unknown. We have discussed that defective TG re-utilization from lipid droplets by lipophagy likely contributes to the accumulation (Line 492 pp). The accumulation of TG is likely caused by influx plus failure of lipolysis and lipophagy. We have now added a discussion of neuron-to-astroglia and vice versa transfer of lipids, that is mostly considered beneficial but also toxic under pathological conditions (Line 446 pp).
It is not known if the enzymatic capacity of triglyceride lipases is impaired after TBI. The expression is mostly low in the brain, but adipose triglyceride lipase (ATGL/PNPLA2) has been shown to be expressed in the brain particularly at the brain to CSF interface and at the blood-brain barrier [4]. In addition, very recently, hormone sensitive lipase (LIPE) was detected at synapses and knockout of LIPE impaired memory functions in mice [5]. To address recent insight to TG lipolysis in the brain we have now added some sentences about triglyceride lipases (Line 504 pp).
Minor points:
Responses below together
Line 86: “consisting in anesthesia, stereotactic mounting,…” should be “consisting of…”
Line 107-108: please explain why one animal was “dropped out.” Was it due to deviation from the protocol, a surgical issue, illness…?
Line 140: use euthanized instead of sacrificed.
Line 141: what temperature was the blood centrifuged at?
Line 210: spelling error- group-vise should be group-wise.
Line 234 and henceforward: please be consistent in whether you use capitalization or not for Y- and X-axis. The study has them as both lower and upper case (e.g., x-axis and X-axis). Please be consistent. There is no reason I am aware of for capitalizing x or the y.
Line 324-326: first sentence needs a reference.
Line 371-372: I think the authors mean forgetful not forgettable. Please change.
Line 436: can the authors explain what they mean by “environmental tissue”?
Line 463: add the word “injury” before phenotype.
We thank the reviewer for pointing out errors of word uses and missing information. They are now corrected/added.
One mouse dropped out during surgery. The information is now added.
We now use "euthanized".
Blood was centrifuged at 4°C. The information is added in the Methods.
It is now x-axis and y-axis without capital letter throughout.
References were added to Line 324-26 addressing endocannabinoid biosynthesis.
Yes, forgetful was meant. It is now corrected.
Line 436 is now reworded.
References
- Cole, J.T.; Yarnell, A.; Kean, W.S.; Gold, E.; Lewis, B.; Ren, M.; McMullen, D.C.; Jacobowitz, D.M.; Pollard, H.B.; O'Neill, J.T., et al. Craniotomy: True sham for traumatic brain injury, or a sham of a sham? J. Neurotrauma 2011, 28, 359-369.
- Krämer, T.; Grob, T.; Menzel, L.; Hirnet, T.; Griemert, E.; Radyushkin, K.; Thal, S.C.; Methner, A.; Schaefer, M.K.E. Dimethyl fumarate treatment after traumatic brain injury prevents depletion of antioxidative brain glutathione and confers neuroprotection. J. Neurochem. 2017, 143, 523-533.
- Vogel, A.; Wilken-Schmitz, A.; Hummel, R.; Lang, M.; Gurke, R.; Schreiber, Y.; Schäfer, M.K.E.; Tegeder, I. Low brain endocannabinoids associated with persistent non-goal directed nighttime hyperactivity after traumatic brain injury in mice. Sci. Rep. 2020, 10, 14929.
- Etschmaier, K.; Becker, T.; Eichmann, T.O.; Schweinzer, C.; Scholler, M.; Tam-Amersdorfer, C.; Poeckl, M.; Schuligoi, R.; Kober, A.; Chirackal Manavalan, A.P., et al. Adipose triglyceride lipase affects triacylglycerol metabolism at brain barriers. J. Neurochem. 2011, 119, 1016-1028.
- Skoug, C.; Holm, C.; Duarte, J.M.N. Hormone-sensitive lipase is localized at synapses and is necessary for normal memory functioning in mice. J. Lipid Res. 2022, 100195.
Reviewer 3 Report
Dear authors,
The manuscript entitled " Phosphatidylethanolamine deficiency and triglyceride over-load in perilesional cortex contribute to non-goal directed hyperactivity after traumatic brain injury in mice” describes y lipids that are deregulated in TBI brain and/or plasma and are associated with behavioral deficits. It presents scientific relevance for the area of Medicine, Cell Biology and Pathology area.
After consulting www.sciencedirect.com and https://pubmed.ncbi.nlm.nih.gov/, publications were found for some authors involving the theme. However, you need to change some details:
-line 84: ."..using the controlled cortical impact (CCI) method" - Was this protocol based on any references? If yes, add!
-line 116: Remove Space
-line 154: " 20,000 g at room temperature" ?
Congratulations, an interesting research article. Good results, with pertinent explications.
Author Response
Dear authors,
The manuscript entitled " Phosphatidylethanolamine deficiency and triglyceride over-load in perilesional cortex contribute to non-goal directed hyperactivity after traumatic brain injury in mice” describes y lipids that are deregulated in TBI brain and/or plasma and are associated with behavioral deficits. It presents scientific relevance for the area of Medicine, Cell Biology and Pathology area.
We thank you for evaluation of our manuscript and your comments that helped us to improve the manuscript.
The changes in the manuscript are highlighted with WORD's track changes. Yellow marked text is rewritten on Editorial request to avoid similarities with own previous work.
After consulting www.sciencedirect.com and https://pubmed.ncbi.nlm.nih.gov/, publications were found for some authors involving the theme. However, you need to change some details:
-line 84: ."..using the controlled cortical impact (CCI) method" - Was this protocol based on any references? If yes, add!
We have added references for the CCI model where they were missing.
-line 116: Remove Space
The figure legend is formatted according to Journal style.
-line 154: " 20,000 g at room temperature" ?
It was ambient temperature. The information is now added.
Congratulations, an interesting research article. Good results, with pertinent explications.
Thank you for your positive evaluation. We much appreciate your opinion.
Round 2
Reviewer 1 Report
The authors have adequately addressed my concerns
Reviewer 2 Report
The authors have addressed my concerns and suggestions adequately in the revised manuscript.